# Evolutionary Origin, Genetic Recombination, and Phylogeography of Porcine Kobuvirus

**DOI:** 10.3390/v15010240

**Published:** 2023-01-14

**Authors:** Yongqiu Cui, Jingyi Li, Jinshuo Guo, Yang Pan, Xinxin Tong, Changzhe Liu, Dedong Wang, Weiyin Xu, Yongyan Shi, Ying Ji, Yonghui Qiu, Xiaoyu Yang, Lei Hou, Jianwei Zhou, Xufei Feng, Yong Wang, Jue Liu

**Affiliations:** 1College of Veterinary Medicine, Yangzhou University, Yangzhou 225009, China; 2Jiangsu Co-Innovation Center for Prevention and Control of Important Animal Infectious Diseases and Zoonoses, Yangzhou University, Yangzhou 225009, China; 3College of Animal Science and Technology, Anhui Agricultural University, Hefei 230036, China

**Keywords:** porcine kobuvirus, phylodynamic analyses, evolutionary origin, genetic recombination, phylogeography

## Abstract

The newly identified porcine Kobuvirus (PKV) has raised concerns owing to its association with diarrheal symptom in pigs worldwide. The process involving the emergence and global spread of PKV remains largely unknown. Here, the origin, genetic diversity, and geographic distribution of PKV were determined based on the available PKV sequence information. PKV might be derived from the rabbit Kobuvirus and sheep were an important intermediate host. The most recent ancestor of PKV could be traced back to 1975. Two major clades are identified, PKVa and PKVb, and recombination events increase PKV genetic diversity. Cross-species transmission of PKV might be linked to interspecies conserved amino acids at 13–17 and 25–40 residue motifs of Kobuvirus VP1 proteins. Phylogeographic analysis showed that Spain was the most likely location of PKV origin, which then spread to pig-rearing countries in Asia, Africa, and Europe. Within China, the Hubei province was identified as a primary hub of PKV, transmitting to the east, southwest, and northeast regions of the country. Taken together, our findings have important implications for understanding the evolutionary origin, genetic recombination, and geographic distribution of PKV thereby facilitating the design of preventive and containment measures to combat PKV infection.

## 1. Introduction

Ongoing epidemiological surveillance is essential for the prevention and control of emerging infectious diseases [1]. According to the International Committee on Taxonomy of Viruses (ICTV, https://talk.ictvonline.org/taxonomy/, accessed on 1 September 2022), Kobuvirus belongs to the Picornaviridae family and comprises six species (A to F). Viruses of species A have been found in humans and canines [2,3,4], viruses from species C have been found in swine [5], and viruses from species B, D, E, and F have been found in cattle, sheep, rabbits, and bats [6,7,8,9,10,11]. Porcine Kobuvirus (PKV) was first identified in healthy and diarrheal swine herds in Hungary and China in 2007 [12,13], after which it was detected in Japan, South Korea, the USA, the Czech Republic, East Africa, and Thailand within a short period of time [14,15,16,17,18,19,20]. The positive rate of PKV in diarrheal pigs was 84.2% and the positive rate in the feces of healthy pigs was 32% in China [21]; however, most of the PKV positive samples from diarrheal pigs were co-infected with one or more of other kinds of pathogens, such as porcine epidemic diarrhea virus, transmissible gastroenteritis virus, pseudorabies virus, and/or Escherichia coli. Kobuvirus is usually transmitted by the fecal-oral route, and infects the gastrointestinal tract [22], indicating that PKV may be closely related to piglet diarrhea. A previous study also proved that PKV can induce diarrhea and specific pathological lesions, including interstitial pneumonia, nephrosis, and gastroenteritis, in piglets [5]. Piglets infected with PKV exhibited a large quantity of defluvium bronchial epithelial cells in the pulmonary artery, a few defluvium alveolar epithelial cells exuded to the alveoli, and the presence of lymph node-like cells, while the pulmonary interstitial tissue appeared to be thickened and congested [5]. The kidney of the piglets showed cellular cast formation with a large number of red blood cells, and several renal tubular epithelial cells exuded in the kidney tubules [5]. The digestive system of the piglets that experienced diarrhea showed marked extravasated blood from the stomach, with few lymphocytes and mononuclear phagocytes infiltrating the submucosa [5].

PKV is a non-enveloped, single-stranded, positive-stranded RNA virus that contains only one open reading frame (ORF) in its genome, which encodes three structural proteins (VP0, VP3, and VP1) and eight nonstructural proteins (L, 2A, 2B, 2C, 3A, 3B, 3C, and 3D) [22]. Mutation, recombination, selection, gene flow, and genetic drift are the most important mechanisms involved in viral evolution. Viruses within the Picornaviridae family often undergo mutation and recombination events [23,24,25]. A previous study has shown that there is a recombination event in PKV [26]; therefore, recombination might play an important role in the evolution of PKV. In recent years, Kobuviruses have been found in other animals, including bats, canines, felines, and goats, they are quite different from previous Kobuviruses [3,11,27]. An epidemiological survey of Kobuviruses in Vietnam found frequent cross-species transmission of Kobuviruses both within and among mammalian species [11].

Currently, large amounts of infectious diseases, including (re)emerging diseases, are becoming increasingly damaging to livestock and poultry in China [28,29,30,31,32]. China has the largest swine population in the world. Swine transportation between various provinces in China is very frequent, and whether this frequent hog trade accelerated the prevalence and variation of swine infectious diseases is unclear [33,34]. Phylogeographic analysis enables the construction of the geographic distribution of a virus through time, based on the sampling times and sampling locations associated with the collected sequence data [28,35,36]. Live animal trade has been shown to be an important source of viral spread in previous phylogenetic studies. For example, phylogeographic inference identified rapid population expansion following the unidirectional spread of the type 2 porcine reproductive and respiratory syndrome virus (PRRSV) from Canada to North America through the hog trade [37]. Swine influenza A virus has been proven to spread through the international live-pig trade [33]. However, whether this type of trade has an influence on spread of the Kobuviruses is still no clear.

In the present study, we constructed an initial dataset including new sequences of four strains of PKV collected in the past 5 years by our group and all the available sequences in GenBank. After rigorous data screening of the sequences, the dataset was analyzed by neighbor-joining (NJ), maximum-likelihood (ML), and Bayesian Inference (BI) methods to study the molecular genetic relationship. We used Bayesian stochastic variable selection (BSSVS) analysis to trace the origin of PRV based on all the available complete coding sequences of Kobuviruses. BSSVS analysis was also used to construct the PKV spread route within China and across the world. Overall, our results provide a global view of the origin, evolutionary dynamics, and phylogeography of PKV and indicate genetic recombination supporting the ongoing genotype shift and outbreaks.

## 2. Methods and Materials

### 2.1. Sequence Datasets, Alignment, and Model Test

PKV datasets: A total of 55 complete genome sequences and 29 complete coding sequences were collected for further analysis. Noncoding regions (5′-UTR and 3′-UTR) were deleted, forming a complete coding dataset 1.

Sequence alignment: Multiple sequence alignments were constructed using multiple alignments performed via the fast Fourier transform (MAFFT) algorithm, and manual checks and sequences were also performed [38]. The final dataset included 80 sequences that we will use in subsequent studies.

Model Test: The best substitution model was selected based on BIC scores calculated using ModelFinder [39]. References suggest that molecular clocks, population dynamics models, and discrete trait substitution models were selected for evaluation of BF values [40].

### 2.2. Recombination Detection

Recombination events occurring in viral genomes negatively impact viral phylogenetic reconstruction. Previously, there was evidence of recombination involving PKV genomes [26]. For recombination detection, we used SplitsTree software, which can conduct phi tests for recombination [41], and RDP 4.0 Beta software, which provides eight methods for recombination detection to further guide removal of recombination sequences [42]. For our analysis, phi detection was performed, and only when the *p* value was <0.05, the seven methods (RDP, CHIMAERA, 3SEQ, MAXCHI, GENECONV, BootScan, and SiScan) in the RDP software were used for detection [43,44,45,46,47,48,49]. After deletion of recombinant strains and recombination regions, dataset 2 was formed for further analysis.

### 2.3. Phylogeographic Analysis

For accurate classification of PKV strains, three distinct methods were used: BI tree using BEAST (1.10.4) [50], ML tree using IQ-TREE [51], and NJ tree using MEGA 7.0 [52]. The final phylogenetic tree was constructed using FigTree software (v.1.4.4) (https://github.com/rambaut/figtree/releases, accessed on 1 September 2022) and iTOL (https://itol.embl.de, accessed on 1 September 2022).

To better analyze the geographical distribution of PKV, we divided PKV into Chinese and global strains, forming dataset 3. To construct the spatial dispersal process, we employed a continuous-time Markov chain (CTMC) process in the BEAST V.1.10.4 package to measure the instantaneous rate of transitions between different regions [35,53]. Furthermore, Bayesian stochastic variable selection (BSSVS) was employed for spatial propagation [35]. To better evaluate prediction results, we introduced the BF cut-off to measure: a BF > 10 indicates very strong support, a BF > 5 means strong support, and a BF > 3 means positive support for PKV spread.

### 2.4. Structural Simulation of PKV VP1 Proteins

VP1 protein structures of different Kobuvirus species were built by homology modeling, and suitable templates were selected for these protein sequences using Alphafold2 for homology modeling [54]. The VP1 protein structure (5 gka) of the human Kobuvirus was downloaded from the RCSB PDB bank (https://www.rcsb.org/, accessed on 1 September 2022). PyMOL was used to generate model images.

## 3. Results

### 3.1. Global Porcine Kobuvirus Epidemic

To understand the prevalence of PKV, we downloaded the complete genome sequences of all available PKV strains from NCBI (Appendix A). After sorting and aligning the sequences, we found that the nucleotide similarity of the complete genomes ranged from 85.2% to 90.8%, and the amino acid similarity of the complete coding sequences ranged from 81.5% to 87.22%. We also used NJ and ML algorithms based on the VP1 gene and the maximum clade credibility (MCC) algorithm based on a complete coding sequence to construct PKV phylogeny. Two independent clades were observed using the three algorithms, all of which revealed similar structures (Figure 1A,B and Figure 5A). Considering the stable structures of the phylogenetic results, we designated them clades PKVa and PKVb. By phylogenetic analysis, we observed that the distribution of a lineage was random in different phylogenetic trees, and we referred to these lineages as intermediate strains (PKV-IM). Moreover, PKVa was separated into three individual clades (PKVa-1, PKVa-2, and PKVa-3), and PKVb was separated into two individual clades (PKVb-1 and PKVb-2) with stable structures (Figure 1B). However, the MCC tree using the complete coding sequence did not display clear clusters, and the PKV-IM strains were clustered into PKVa and PKVb.

### 3.2. Genetic Recombination Involving Porcine Kobuvirus

Recombination of viral genomes is known to negatively impact the phylogenetic reconstruction of Kobuvirus as a single-stranded positive-stranded RNA virus, which has been demonstrated to have a high substitution rate in the range of 10^−3^ [55]. In this study, we used all full genome sequences for genetic recombination analysis using SplitsTree and RDP 4.0 Beta software. More than half of these genome sequences were identified as being recombinant, with frequent recombination fragments involving nucleotide positions 2762–3759 (Figure 2A). After that, we used a recombinant fragment to perform ML phylogenetic reconstruction according to positions 2762–3759 (encoding VP3 + VP1 + 2A) (Figure 2B). Combined BootScan analysis and phylogenetic reconstruction results indicated that Group A evolved from a recombinant virus that acquired the fragment 2762–3759 of VP3, VP1, and 2A from MT125684; the other genetic recombinants identified are shown in Figure 2a. The above results indicated that recombination played an important role in the evolution of PKV.

### 3.3. Relationship between Multiple Species of Kobuviruses

To trace the origin of PKV and its relationship to other species of Kobuviruses, Bayesian stochastic variable selection (BSSVS) analysis was performed using a complete coding sequence that included deletion of recombination regions and recombination sequences. We found that all PKV strains were closely related to the bovine Kobuvirus (BKV) with high confidence (Figure 3). Additionally, after labeling the node’s posterior probabilities as ancestral animals, we found that Kobuvirus was mainly divided into two groups, and the most recent ancestor of both groups was the rabbit Kobuvirus, sheep might be an intermediate host in the spread of Kobuvirus. The BSSVS analysis results indicated that bat Kobuvirus has close relationship with human Kobuvirus, and human Kobuvirus showed high confidence with canine and feline Kobuvirus (Figure 3). In addition, the Kobuvirus (MT766372) found in sewage shared 89.8–99.7% homology to human Kobuvirus, which indicated that Kobuvirus may be transmitted through fecal-oral route (Yang et al. 2021). We also calculated the divergence time of Kobuvirus based on currently available sequence information. The results showed that Kobuvirus is an ancient virus that can be traced back to around 1500, and PKV infection serves as an emerging infectious disease, the divergence time of which can be traced back to around 1900 (Figure 3), which has been circulating for a long period of time before being detected.

Binding of viruses to cell surface receptors is a prerequisite for viruses to infect cells, and this is also a prerequisite for the cross-species transmission of viruses. The VP1 protein is an important component of the viral capsid, and it is also a key protein that binds to cell surface receptors. We compared the homology of VP1 proteins of different species of Kobuvirus and found that homology at the amino acid level was not high, but the amino acids at the 13–17 and 25–40 residue motifs of the PKV VP1 protein showed high homology of >95%, except for rabbit and bat Kobuviruses (Appendix A). To better determine the position of these two highly conserved motifs in the structure of the VP1 protein, we used AlphaFold2 to simulate the structure of the Kobuvirus VP1 protein (Figure 4 and Appendix A). Interestingly, we found that both these motifs were exposed on the exterior of the viral capsid surface (Figure 4A,B). When the surfaces of different species of Kobuviruses have similar structures and are exposed on the outside, they may bind to the same or similar receptors on the cell surface of different species, thus leading to the transmission of Kobuviruses within different species.

### 3.4. Phylogeographic Construction of Global Spread Route of Porcine Kobuvirus

Constructing a model of the spread of a virus over time is subject to many uncertainties, which are mainly reflected in observed lower posterior probabilities. Therefore, accounting for this uncertainty, relationships among strains were estimated in terms of well-supported (Bayesian factor (BF) > 3, Posterior Probability (PP > 0.3)) contacts among countries.

Except for a certain inter-dataset variability, the overall results supported the presence of viral spreading, corresponding to America, Europe, Asia, and Africa; Europe might be an origin for the spread of PKV to Asia and Africa, with Germany acting as an intermediary between Europe and Africa (Figure 5 and Table 1). Within these major continents, several local migration routes were demonstrated to be significant based on current strain information, with Spain being the region most likely to drive the spread of PKV to other countries, including China, Japan, Germany, South Africa, Hungary, and the Netherlands; whereas the route of PKV spread within Europe, America, and Asia is still unclear. Although PKV has been reported in two American countries, the United States of America and Mexico, the PKV strains in the United States were more likely mediated by viral strain exchange with China than Mexico strains. Meanwhile, a Bayesian skygrid plot was constructed to reflect changes in the effective PKV population size (Appendix A); we found that the effective population of PKV remained at a relatively stable size before 2008, decreased after 2008, decreased again around 2012, and then remained at a lower group level until recently.

### 3.5. Phylogeographic Reconstruction of the Spread Route of Porcine Kobuvirus within China

We also constructed geographic dispersal and explored the effective population change of PKV within China. Phylogeographic construction analysis revealed that long-distance movement of PKV between different provinces within China has occurred continuously in swine herds since 1995 (Figure 6A and Table 2). Figure 6A illustrates the construction transmission of PKV in China over the past three decades based on discrete phylogenetic inference using a generalized linear model (GLM). Construction analysis indicated that Hubei Province was a possible origin with a high posterior support of 0.9811. Shortly after the first occurrence in mainland China, some strains from Jiangsu, Anhui, Gansu, and Guangdong were all originated from Hubei PKV strain, which all had high posterior support. Subsequently, with the transfer of live pigs across several provinces and cities, Jiangsu, Gansu, and Jiangxi gradually became centers of PKV transmission and spread to Shanghai, Henan, Jilin, Heilongjiang, and other provinces (Figure 6B). From its inferred time of origin, the differentiation of PKV was mainly concentrated between 1995–2005, and a high number of effective populations were maintained throughout the decade (Figure 6A and Appendix A). However, the effective population of PKV declined twice, in 2010 and 2018 (Appendix A), which may also be related to the decline in domestic pig herds caused by reemerging swine virus outbreaks.

## 4. Discussion

The prevention and control of emerging infectious diseases in China and across the world depends on a deep understanding of how to quickly we can detect emerging infectious diseases and model transmission across geographic and national boundaries. The newly identified PKV, due to its potential harm to the pig-rearing industry, has led to great interest in the epidemiology and biology of PKV. One of the most important questions to be answered regarding the virus is the origin and spread of PKV. As an increasing number of pig viruses have been discovered in recent years [56], a new question has also been raised: Have these viruses been circulating among pigs for a prolonged period or just only recently been discovered due to advanced detection technology and methods? Here, for the first time, we provide new insights into the origin, epidemiology, and transmission routes of PKV within China and across the world.

Accurate viral typing is helpful to better understand variations in the virus epidemic. Considering that VP1 is the gene with the highest degree of variation in the PKV genome, and the previously data that canine Kobuvirus and other Kobuvirus genotypes were identified based on VP1 gene sequences [56,57], we constructed the phylogenetic history of PKV based on all the available strain sequences. For PKV genotyping, two different methods, NJ and ML, were used, and two stable clades (PKVa and PKVb) and one intermediate clade (PKV-IM) were identified (Figure 1a). Additionally, within the PKVa and PKVb clades, three (PKVa-1, -2, and -3) and two (PKVb-1 and PKVb-2) subclades were identified, respectively (Figure 1b). The phylogenetic classification results showed that the genetic diversity of PKVa is higher than that of PKVb, and the divergence time of PKVa is earlier than that of PKVb. However, PKVb is currently the most prevalent strain, and the prevalence of PKVb-1 was similar to that of PKVb-2. PKVb may have resulted from long-term evolutionary selection, and PKVa strains have gradually become vulnerable. It is reasonable to use the ML method to classify the PKV VP1 gene, which helps us to better understand epidemic trends involving PKV and to propose further control measures for epidemic strains in the future.

Genetic recombination is an important factor for viral evolution [58,59]. During replication, a set of subgenomic RNAs is generated, increasing the homologous recombination rate among closely related genes from different lineages of Kobuvirus or other viruses by template switching [60,61]. To detect genetic recombination involving PKV, we first used phi tests in SplitsTree software and revealed a very strong recombination signal. Then, we used RDP software to detect specific recombinant strains and recombination regions (Figure 2A). More than 60 PKV strains were found to be recombinant, with the recombination region located at 2762–3759 nt; this recombinant region encodes the VP3 protein and part of the VP1 and 2A proteins. Following the discovery of this recombination event, we performed a phylogenetic analysis using the recombination fragments and found that PKV was clearly divided into two groups, and all recombinant strains formed Group A (Figure 2B). These results demonstrated that PKV is likely to have a high recombination frequency, which generates novel viruses with high genetic diversity. Meanwhile, recombination increases genetic diversity and the likelihood of adaptation to new hosts [61,62]. As an RNA virus, PKV may have a high probability of recombination owing to the random switching of templates during RNA replication. This mechanism may be similar to the coronavirus recombination mechanism mediated by the “copy choice” mechanism of viral polymerases [63]. There are multiple hog production systems in the world, such that hogs can be traded through multiple channels [64], and that multiple channels can creates contacts between different PKV lineages, which may be another possibility involving higher recombination rates. Notably, our results suggested that recombination plays an important role in the evolution of PKV, which deserves further attention in the future.

To further understand the origin of PKV, we used the BSSVS method to analyze different species of Kobuvirus, including human, rabbit, sheep, bovine, canine, feline, and swine. We found that Kobuvirus can be mainly divided into two large groups: one group mainly includes bats, humans, canines, and felines, and the other large group includes pigs, cattle, and sheep. In addition, we identified that rabbit Kobuvirus was the most recent common ancestor of Kobuvirus. (Figure 3). After labeling the posterior probability of each node as the ancestral host, we found that the SKV showed a close relationship with PKV and BKV (Figure 3). Swine, bovine, and sheep are the most widespread economically important animals, and are often mixed on farms. In addition, Kobuvirus may be transmitted through fecal-oral transmission; therefore, it may cause continuous transmission of the virus between porcine, bovine, and sheep. Bats are generally considered to be important hosts for novel emerging infectious diseases, such as influenza virus, Marburg virus, Nipha virus, Hendra virus, coronavirus, and Ebola virus [65,66,67,68,69,70]. Furthermore, it is well known that most human pathogens originate in animals and arise through cross-species transmission [71]. Furthermore, the analysis results suggest that cross-species transmission of Kobuviruses exists. As companion animals, canines and felines have very close contact with humans, and canine and feline Kobuvirus can be detected in feces [3,72], which makes it possible for the virus to spread to canines and felines through fecal-oral transmission. To further understand the relationship between different species of Kobuvirus, after analyzing the amino acid sequence and structure of the VP1 protein, we found that there are two motifs with >95% homology in mammalian Kobuviruses, except for rabbit and bat Kobuviruses (Appendix A), and both motifs are exposed on the exterior of the VP1 protein (Figure 4 and Appendix A), suggesting that these two motifs may be key points for Kobuvirus to infect different mammalian cells. The basis for cross-species transmission of Kobuvirus should be based on the fact that highly conserved motifs (sequences and structure) of the viral VP1 protein can recognize the same receptor in different hosts. The receptor-binding domain of VP1 should be the determination of host range of Kobuvirus. However, in terms of genetic similarity, rabbit Kobuvirus and bat Kobuvirus are quite different from other Kobuviruses. We consider that rabbits may not directly transmit Kobuvirus to other animals in the long-term evolution, and there may be intermediate hosts that we have not yet discovered. Combining these analysis results, we can answer the question that PKV has persisted for a prolonged period, and could not been detected until recent years when the diagnostic technology had advanced.

In this study, we constructed the spread routes of PKV using phylogeographic methods based on high-quality and updated sequence datasets with sampling times longer than three decades. Despite our best efforts, currently available data are limited and may vary owing to inconsistent testing and sequencing methods in different countries. Based on our phylogeographic analysis results (Table 1), the PKV origin was dated before 1975, and Spain was the region most likely to drive the spread of PKV to other countries (Figure 5). However, on a global scale, Asia (China and Japan) and Europe (Spain and Germany) maintained the highest numbers of PKV lineages (Figure 5). We estimated that after an initial introduction out of Spain, PKV may have spread to other countries (China, Japan, the Netherlands, Germany, Hungary, and South Africa) through trade or by other means (e.g., intermediate host) of transmission. Most Asian and European introductory events came from North America, which indicated that greater attention should be paid to the export of live pigs and pig-related products from North America to Asia and Europe. Meanwhile, we observed that China was not a major exporter of live swine, which was also reflected in our analysis results; we found only a small number of PKV spread events from China to America. We also constructed a Bayesian skygrid tree prior to measuring the dynamic changes in demographic history (Appendix A). Interestingly, we found two sharp declines in the effective populations of PKV in the 2 years before and after 2010 that may be related to an outbreak of swine influenza virus around 2010 [73], which caused a decline in the number of pig herds and cross-border pig transfers.

China is currently the country with the highest numbers of live pigs in the world and is one of the countries with frequently emerging infectious diseases. Therefore, it is necessary to study the spread of PKV in China. We observed that the spread of PKV within China was strongly promoted by the swine trade (swine transport and products) within provinces, with Hubei, Gansu, Jiangsu, and Jilin provinces representing the main hubs of viral spread and seeding of PKV into other provinces, including Anhui, Guangdong, Jiangxi, Henan Province, and Shanghai City (Figure 6A, Table 2). Based on our results, Hubei Province is the center of PKV spread in China (Figure 6B). Combining the perspective of geographical distribution, Hubei Province is also the most central province in China and has very convenient transportation channels, which greatly helps PKV spread to the east, southwest, and northeast regions of China. Interestingly, the spread route of PKV is consistent with that of porcine epidemic diarrhea virus (PEDV) in China [28]. It is worth noting that pig feed contamination and pork consumption (causing vehicle contamination and slaughterhouse contamination) may also be possible factors in PKV spread, which has been demonstrated in PEDV [28]. Our study accurately constructed early spread routes of PKV in China. We found that PKV spread from the Hubei Province to southwest (Gansu Province) and northeast (Jilin province) China in the early stages, and this early long-distance transmission was most likely caused by the transportation of live pigs. Based on our Bayesian skygrid results, we found that the effective population of PKV in China decreased twice in 2010 and 2018, which may be due to an outbreak of swine influenza A and African swine fever viruses in China in 2010 and 2018, respectively [8,73,74]. Our findings will provide further insights into controlling the spread of viral diseases in China while helping to identify key areas for prevention and control.

## Figures and Tables

**Figure 1 viruses-15-00240-f001:**
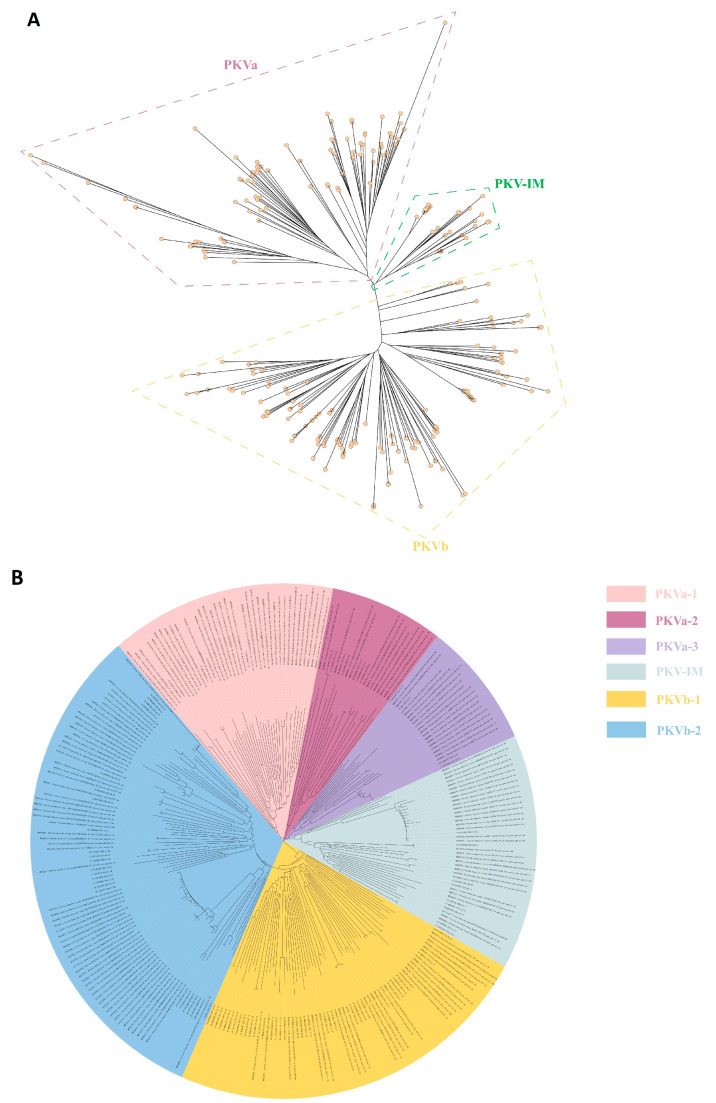
Phylogenetic analysis of all the available 277 VP1 genes sequences. (**A**). NJ tree reconstructed using MEGA 7.0, and different clades are framed by dotted lines of different colors. (**B**). ML tree reconstructed using IQ-TREE, and different clades are marked with different colors.

**Figure 2 viruses-15-00240-f002:**
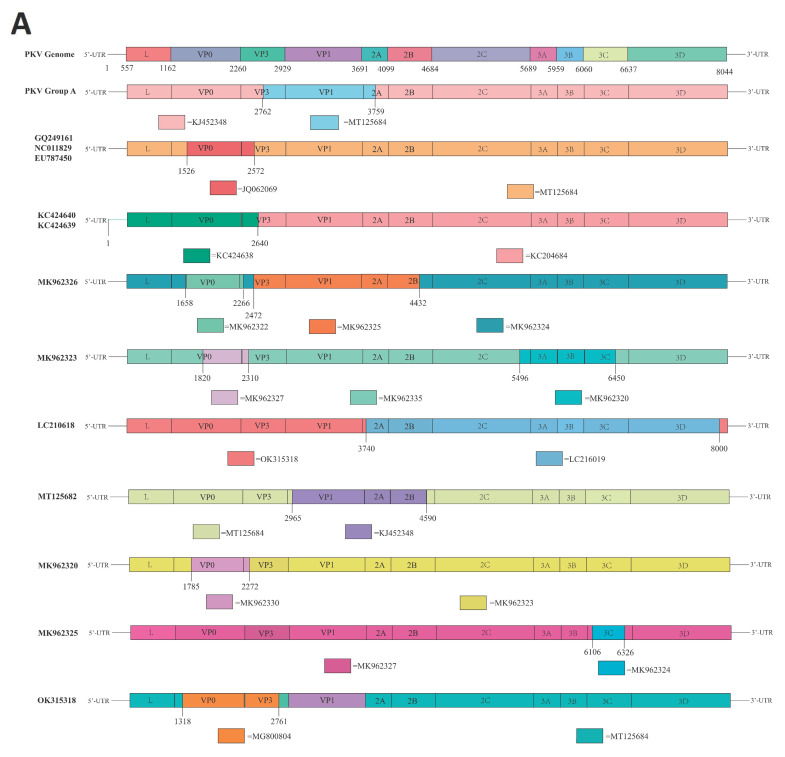
Recombination events of PKV. (**A**). The colored regions of each PKV indicate genetic recombination at the genomic sites with other Kobuvirus origin, and the location of the breakpoint is also marked. (**B**). Phylogenetic analysis of recombination regions based on the ML method; two different groups are marked with two different colors.

**Figure 3 viruses-15-00240-f003:**
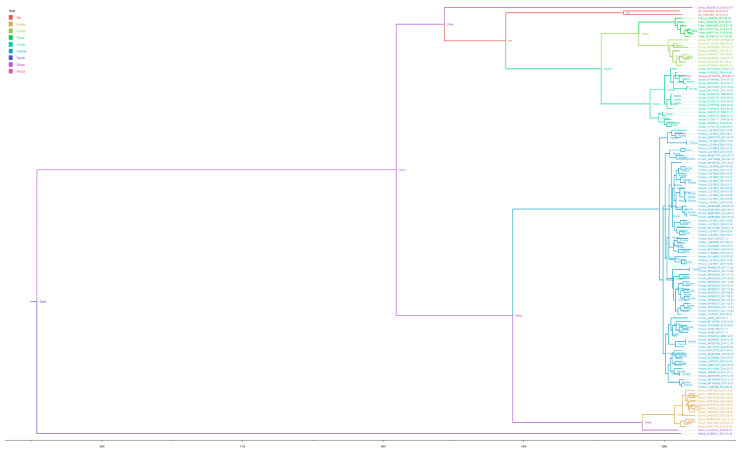
The origin of PKV was deduced using the complete coding region after removal of recombination region. BI tree reconstructed using BSSVS algorithm in BEAST (Version 1.10.4). Lines with different colors represent different Kobuvirus species, bat Kobuvirus (red lines), bovine Kobuvirus (yellow lines), canine Kobuvirus (reseda lines), feline Kobuvirus (bottle green lines), human Kobuvirus (mint green lines), PKV (blue lines), rabbit Kobuvirus (mazarine lines), sheep Kobuvirs (purple lines), and swage Kobuvirus (amaranth lines). The posterior probabilities of different clades are expressed in the most recent common ancestral animals.

**Figure 4 viruses-15-00240-f004:**
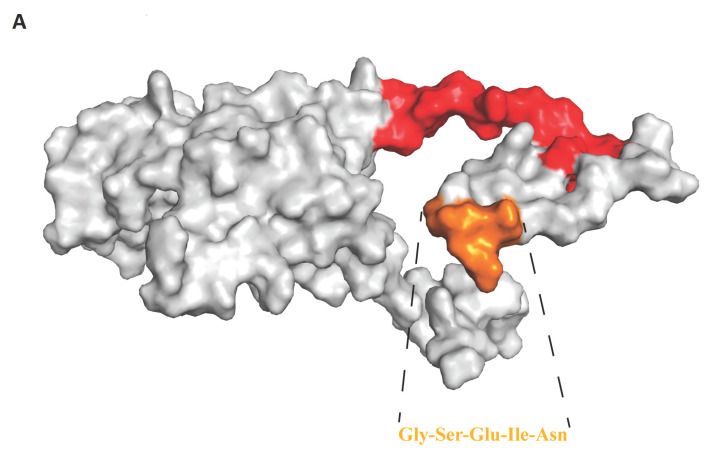
A schematic diagram of the structure simulation of the VP1 protein of PKV using AlphaFold2. (**A**). The orange region represents the motif 13–17 of VP1 protein of the PKV. (**B**). The red region represents the motif 25–40 of VP1 protein of the PKV.

**Figure 5 viruses-15-00240-f005:**
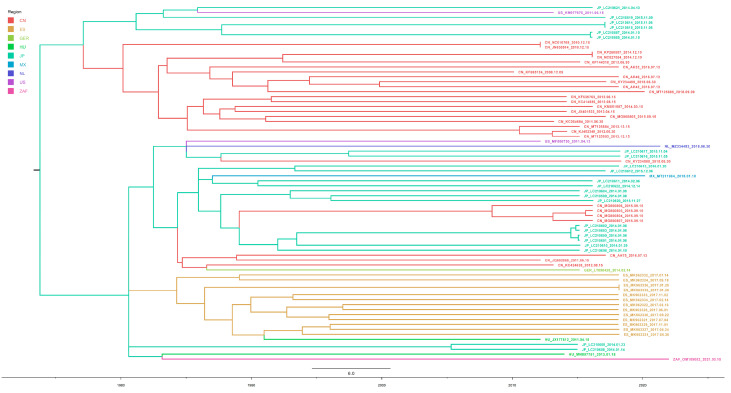
The transmission route of PKV across the world. BI tree of the complete coding region after removal of recombination region is reconstructed using BEAST under the skygrid model.

**Figure 6 viruses-15-00240-f006:**
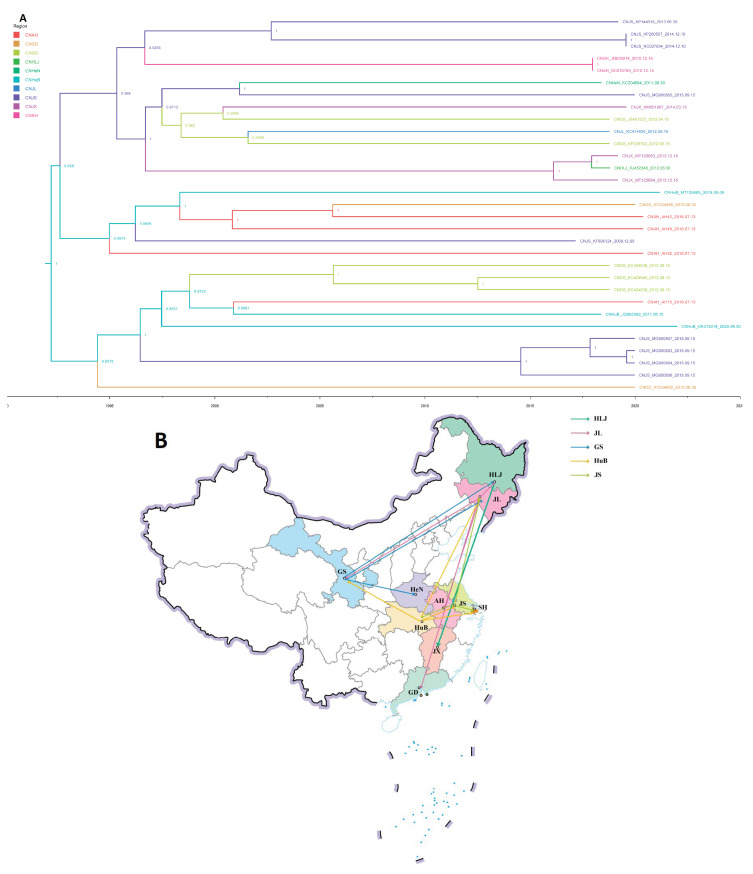
The transmission route of PKV within China. (**A**). Maximum clade credibility tree with annotated posterior probability on the ancestral nodes, depicting the Chinese spread of PKV. Lines with different colors represent different regions, CNAH (red lines), CNGD (yellow lines), CNGS (grass green lines), CNHLJ (reseda lines), CNHeN (bottle green lines), CNHuB (mint green lines), CNJL (blue lines), CNJ (purple lines), CNJX (amaranth lines), and CNSH (rose red lines). (**B**). Reconstruction of PKV transmission route within China.

**Table 1 viruses-15-00240-t001:** Bayes Factor (BF) and Posterior Probability (PP) tests for PKV transmission links across the world.

From	To	Bayes Factory (BF > 3)	Posterior Probability (PP > 0.3)
ES	NL	356.4557507	0.982689274
JP	GER	127.4503405	0.953045488
ES	JP	33.98447793	0.84404791
CN	US	20.69907691	0.767249617
ES	HU	18.4158222	0.74569454
ES	ZAF	8.492634594	0.574920557
ES	GER	5.94957951	0.486522522
ZAF	GER	3.118198076	0.331814849
GER	ZAF	3.042862595	0.326414969

CN: China, ES: Spain, GER: Germany, HU: Hungary, JP: Japan, NL: the Netherlands, ZAF: South Africa, US: The Unite States of America.

**Table 2 viruses-15-00240-t002:** Bayes Factor (BF) and Posterior Probability (PP) tests for PKV transmission links within China.

From	To	Bayes Factory (BF > 3)	Posterior Probability (PP > 0.3)
CNHLJ	CNJX	432.0656717	0.98118565
CNJS	CNSH	15.04857229	0.644934693
CNJL	CNAH	7.108926058	0.461803412
CNJL	CNGS	6.87832112	0.453618429
CNGS	CNHeN	6.747890566	0.448877805
CNGS	CNHLJ	6.64289559	0.445001358
CNHuB	CNJS	6.294587779	0.431742426
CNJL	CNGD	5.342199925	0.392027357
CNJS	CNHuB	4.855507895	0.369509395
CNJL	CNHLJ	4.817537965	0.367682279
CNHuB	CNJL	4.306213326	0.342003901
CNGS	CNJL	3.930574401	0.321769833
CNHuB	CNGS	3.529714165	0.318758055
CNJS	CNJL	3.527426646	0.308622256
CNHuB	CNSH	3.112557463	0.303092022

CNAH: Anhui Province, China, CNHLJ: Heilongjiang Province, China, CNHuB: Hubei Province, China, CNHeN: Henan Province, China, CNGD: Guangdong Province, China, CNGS: Gansu Province, China, CNJS: Jiangsu Province, China, CNJL: Jilin Province, China, CNSH: Shanghai city, China.

## Data Availability

The data supporting the findings of this study are available upon request from the corresponding author.

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
