# Peer review of "Evolutionary Origin, Genetic Recombination, and Phylogeography of Porcine Kobuvirus"

_viruses, 2023, doi:10.3390/v15010240_

Round 1
Reviewer 1 Report
The manuscript of “Evolutionary Origin, Genetic Recombination, and Phylogeography of Porcine Kobuvirus” by Cui described the potential origin and evolution path of the PKV. The results showed that PKV contains two major clades, recombination events occured among different KV strains, which resulted into highly genetic diversity. Further analyis suggested that the PKV might be derived from the rabbit Kobuvirus, and sheep might act as an important intermediate host before infecting pigs.the findings of this study interesting and helpful to understand the evolution of PKU.
Comments
1 Some backgroud informaion of the KV should be included in this article, such as Receptor, replication cycle, and pathgenicity of KV.
2The results suggested that the Cross-species transmission of PKV might be linked to interspecies conserved amino acids at 13–17 and 25–40 residue motifs of Kobuvirus VP1 proteins.The function of the 13–17 and 25–40 residue motifs should be discussed based on the crystal structure model. Whether the different original KV viruses recognize similar receptor of different host? The RBD of VP1 might be the determination of host range of KU.
Author Response
Dear editor and reviewers,
Thank you very much for the chance you gave us to revise our manuscript. We thank the reviewers for the time and effort that they have put into reviewing the previous version of the manuscript. We admire your expertise and patience. The comments are very useful for our research and publication. We decide to accept all the comments and revise our manuscript carefully according to your comments. The changes were listed blew point by point, and were marked in red in the revised manuscript. The manuscript has greatly benefited from these invaluable suggestions.
Based on the instructions provided in your letter, we uploaded the file of the revised manuscript. Appended to this letter is our point-by-point response to the comments raised by the reviewers.
We look forward to working with you to move this manuscript closer to publication in Viruses-Basel.
Reviewer 1
The manuscript of “Evolutionary Origin, Genetic Recombination, and Phylogeography of Porcine Kobuvirus” by Cui described the potential origin and evolution path of the PKV. The results showed that PKV contains two major clades, recombination events occurred among different KV strains, which resulted into highly genetic diversity. Further analysis suggested that the PKV might be derived from the rabbit Kobuvirus, and sheep might act as an important intermediate host before infecting pigs. the findings of this study interesting and helpful to understand the evolution of PKU.
Comments
1 Some background information of the KV should be included in this article, such as Receptor, replication cycle, and pathogenicity of KV.
Response: As a novel infectious disease, PKV has not attracted much attention at present, so there are no reports related to PKV receptors and replication cycle. The information of pathogenicity has been added, as described in detail on Pages 1-2 lines 42-53 of the revised manuscript and the relevant reference has been inserted into Page 17 lines 499-500 of the revised one.
2 The results suggested that the Cross-species transmission of PKV might be linked to interspecies conserved amino acids at 13–17 and 25–40 residue motifs of Kobuvirus VP1 proteins. The function of the 13–17 and 25–40 residue motifs should be discussed based on the crystal structure model. Whether the different original KV viruses recognize similar receptor of different host? The RBD of VP1 might be the determination of host range of KV. Response: The basis for cross-species transmission of KV should be based on the highly conserved motif (sequences and structure) of the viral VP1 protein can recognize the same receptor in different hosts. Thus, different original KV can recognize similar receptor of different host. Structural proteins form the capsid of KV and are associated with adsorption and invasion when KV infects the cell, so, the receptor-binding domain of VP1 should be the determination of host range of KV. The relevant description has been inserted into Page 14 lines 370-373 of the revised manuscript.

Reviewer 2 Report
Lines 77-78: “Markov chain Monte Carlo (MCMC)” is actually only algorithm of Bayesian Inference in phylogenetics
Later in lines 108 authors used MCC as Markov chain Monte Carlo (not MCMC!), I think both is incorrect — use BI, Bayesian Inference instead
Line 109 MEAG — is MEGA
Figure 3 is impossible to read, the legend is too small too
Line 230 I don’t understand what is TMUV. In addition, I don’t remember you mentioned MrBayes in methods.
Line 233: mentioning RGB colors is useless, nobody will associate numbers with real colors, just provide a legend. Whole legend and caption of Fig. 5 is misleading.
Section 3.5 Phylogeographic Construction is actually Phylogeographic Reconstruction
Fig. 6 B Map of mainland China would be enough since you don’t have any localities outside it. 6 A again difficult to read the legend.
The main problem is that while calculating an effective number of the population, authors took a non-random and very limited set of sequences, done some years ago. Because of that, Bayesian skyplot graphs based on genetic diversity among samples are misled by the amount of variation between individual sequences and reflect rather a decrease of whole genome sequences submission to GenBank, not the natural process in the virus population.
Author Response
Dear editor and reviewers,
Thank you very much for the chance you gave us to revise our manuscript. We thank the reviewers for the time and effort that they have put into reviewing the previous version of the manuscript. We admire your expertise and patience. The comments are very useful for our research and publication. We decide to accept all the comments and revise our manuscript carefully according to your comments. The changes were listed blew point by point, and were marked in red in the revised manuscript. The manuscript has greatly benefited from these invaluable suggestions.
Based on the instructions provided in your letter, we uploaded the file of the revised manuscript. Appended to this letter is our point-by-point response to the comments raised by the reviewers.
We look forward to working with you to move this manuscript closer to publication in Viruses-Basel.
Reviewer 2
Lines 77-78: “Markov chain Monte Carlo (MCMC)” is actually only algorithm of Bayesian Inference in phylogenetics
Response: The modification has been made on Page 2 line 83 of the revised manuscript.
Later in lines 108 authors used MCC as Markov chain Monte Carlo (not MCMC!), I think both is incorrect — use BI, Bayesian Inference instead
Response: The modification has been made on Page 3 lines 114-115 of the revised manuscript.
Line 109 MEAG — is MEGA
Response: The modification has been made on Page 3 line 115 of the revised manuscript.
Figure 3 is impossible to read, the legend is too small too
Response: The Figure 3 has been reuploaded, you can see it in the revised manuscript.
Line 230 I don’t understand what is TMUV. In addition, I don’t remember you mentioned MrBayes in methods.
Response: The wrong statements has been deleted, and the modification has been made on Page 10 lines 258-261 of the revised manuscript.
Line 233: mentioning RGB colors is useless, nobody will associate numbers with real colors, just provide a legend. Whole legend and caption of Fig. 5 is misleading.
Response: The modification has been made on Page 10 lines 258-261 of the revised manuscript.
Section 3.5 Phylogeographic Construction is actually Phylogeographic Reconstruction
Response: The modification has been made on Page 11 line 266 of the revised manuscript.
Fig. 6 B Map of mainland China would be enough since you don’t have any localities outside it. 6 A again difficult to read the legend.
Response: Fig. 6B is a map of the whole territory of China downloaded from the official website of the Ministry of Natural Resources of the People’s Republic of China (https://www.mnr.gov.cn/). Fig. 6A has been reuploaded, and the legend has been modified on Pages 12-13 of lines 286-289 of the revised manuscript.
The main problem is that while calculating an effective number of the population, authors took a non-random and very limited set of sequences, done some years ago. Because of that, Bayesian skyplot graphs based on genetic diversity among samples are misled by the amount of variation between individual sequences and reflect rather a decrease of whole genome sequences submission to GenBank, not the natural process in the virus population.
Response: Thank you for your very important advice. In order to ensure the correctness of the calculation results, we extended the calculation times as much as possible to ensure that the ESS values were within the credible range. However, the analysis will need to be repeated in the future as more viral sequences become available to better reflect the dynamics of the virus population as suggested by the reviewer.

Round 2
Reviewer 2 Report
The authors accepted most of the corrections and did changes which had improved the manuscript except for what seems to be the major flaw - demographic reconstruction is made on an incomplete dataset and could not be improved without including more samples.
Author Response
Dear editor and reviewers,
Thank you very much for the chance you gave us to revise our manuscript. We thank the reviewers for the time and effort that they have put into reviewing the previous version of the manuscript. We admire your expertise and patience. The comments are very useful for our research and publication. We decide to accept all the comments and revise our manuscript carefully according to your comments. The changes were listed blew point by point, and were marked in red in the revised manuscript. The manuscript has greatly benefited from these invaluable suggestions.
Based on the instructions provided in your letter, we uploaded the file of the revised manuscript. Appended to this letter is our point-by-point response to the comments raised by the reviewers.
We look forward to working with you to move this manuscript closer to publication in Viruses-Basel.
Reviewer 2
The authors accepted most of the corrections and did changes which had improved the manuscript except for what seems to be the major flaw - demographic reconstruction is made on an incomplete dataset and could not be improved without including more samples.
Response: Thank you for your very important advice. We have moved this result from the main manuscript to the supplementary materials of the revised manuscript. The modification has been made on Page 9 line 256, Page 11 lines 292-293, and Page 14 lines 426 and 462-466 of the revised manuscript.
